# Sleep and Awakening Quality during COVID-19 Confinement: Complexity and Relevance for Health and Behavior

**DOI:** 10.3390/ijerph18073506

**Published:** 2021-03-28

**Authors:** Teresa Paiva, Cátia Reis, Amélia Feliciano, Hugo Canas-Simião, Maria Augusta Machado, Tânia Gaspar, Gina Tomé, Cátia Branquinho, Maria Raquel Silva, Lúcia Ramiro, Susana Gaspar, Carla Bentes, Francisco Sampaio, Lara Pinho, Conceição Pereira, Alexandra Carreiro, Susana Moreira, Isabel Luzeiro, Joana Pimentel, Gabriela Videira, Júlio Fonseca, Ana Bernarda, Joana Vaz Castro, Sofia Rebocho, Katie Almondes, Helena Canhão, Margarida Gaspar Matos

**Affiliations:** 1Sleep Medicine Center—CENC, 1070-068 Lisbon, Portugal; reis.catia@gmail.com (C.R.); amelia.feliciano71@gmail.com (A.F.); hugo.simiao@gmail.com (H.C.-S.); tania.gaspar.barra@gmail.com (T.G.); joapim@gmail.com (J.P.); joanavazdecastro@gmail.com (J.V.C.); sofia.rebocho@gmail.com (S.R.); 2Instituto de Saúde Ambiental (ISAMB), Faculdade de Medicina, Universidade de Lisboa, 1649-026 Lisbon, Portugal; ginatome@sapo.pt (G.T.); catiasofiabranquinho@gmail.com (C.B.); lisramiro@sapo.pt (L.R.); smsgaspar@gmail.com (S.G.); margarida.gaspardematos@gmail.com (M.G.M.); 3Comprehensive Health Research Center (CHRC), Nova Medical School, Universidade Nova de Lisboa, 1169-056 Lisbon, Portugal; raquel@ufp.edu.pt (M.R.S.); lmgp@uevora.pt (L.P.); helena.canhao@nms.unl.pt (H.C.); 4Faculdade de Medicina de Lisboa, Instituto de Medicina Molecular João Lobo Antunes (IMM), 1649-028 Lisbon, Portugal; ccbentes@gmail.com; 5Católica Research Centre for Psychological Family and Social Wellbeing (CRC-W), Catholic University, 1649-023 Lisbon, Portugal; 6Clínica Lusíadas de Almada/Parque das Nações-Lisboa 2810-500, 1990-095 Almada, Portugal; 7Psychiatry and Mental Health Department—Centro Hospitalar de Lisboa Ocidental (CHLO), 1349-019 Lisbon, Portugal; 8Departamento de Medicina, Serviço de Pneumologia, Centro Hospitalar Póvoa de Varzim/Vila do Conde, 4490-421 Póvoa do Varzim, Portugal; maugustamachado68@gmail.com; 9Centro Lusíada de Investigação em Serviço Social e Intervenção Social (CLISSIS), Universidade Lusíada, 1349-001 Lisboa, Portugal; 10Faculty Human Kynetics, Universidade de Lisboa, 1495-751 Lisboa, Portugal; 11Faculty of Health Sciences/Higher School of Health, University Fernando Pessoa/Fundação Fernando Pessoa, 4200-150 Porto, Portugal; fsampaio@ufp.edu.pt; 12Escola Secundária Poeta Al Berto, 7520-902 Sines, Portugal; 13Escola Superior de Saúde Atlântica, Universidade Atlântica, 2730-036 Barcarena, Portugal; 14Department of Neurology, CHULN—Centro Hospitalar Universitário Lisboa Norte, 1649-028 Lisbon, Portugal; 15Neurology and Dental Medicine, Hospital da Luz, 1500-065 Lisbon, Portugal; gabrielapsv@yahoo.com; 16Innovation & Development in Nursing research group, CINTESIS—Center for Health Technology and Services Research, 4200-450 Porto, Portugal; 17Nursing Department, Escola Superior de Enfermagem S. João de Deus (ESESJD), University Évora, 7000-811 Évora, Portugal; 18Instituto de Sono e Medicina Dentária (ISMD), 9000-098 Funchal, Portugal; conceicaofrancopereira@gmail.com; 19Unidade do Sono, Hospital do Divino Espírito Santo de Ponta Delgada, 9500-370 Ponta Delgada, Portugal; alexandracarreiro@hotmail.com; 20Pneumologia, Hospital dos Lusíadas, 1500-458 Lisbon, Portugal; susanalmoreira@gmail.com; 21Centro Medicina Sono e Serviço Neurologia e Neurofisiologia, Hospital CUF Coimbra, 3000-600 Coimbra, Portugal; isabeluzeiro@gmail.com; 22Serviço Neurologia e Neurofisiologia, Hospital CUF Coimbra, 3000-600 Coimbra, Portugal; 23Escola Superior de Tecnologia da Saúde de Coimbra, 3046-854 Coimbra, Portugal; 24Pneumologia, Hospital de Vila Franca de Xira, 2600-009 Vila Franca de Xira, Portugal; 25Orisclinic—Centro Integrado de Medicina Dentária de Coimbra, 3030 Coimbra, Portugal; jfonsecas@hotmail.com; 26Linde Saúde—Homecare, Linde Saúde, 1800-217 Lisbon, Portugal; ana.bernarda@linde.com; 27UFRN—AMBSONO Sleep Clinic, Department Psychology, Federal University Rio Grande do Norte, Natal 59030-180, Brazil; katie.almondes@gmail.com

**Keywords:** health, COVID-19, sleep/awakening quality, Calamity Experience Check List, mood, attitudes, health-related behaviors, dependences, health professionals, sleep patients

## Abstract

Objective: The aim of this study was to evaluate sleep and awakening quality (SQ and AQ) during COVID-19 in a large and diversified population in order to identify significant associations and risks in terms of demography, health and health-related behaviors, sleep variables, mental health, and attitudes. Methods/Results:Online surveys were used for data collection, received from 5479 individuals from the general population, sleep disorder patients, and COVID-involved (medical doctors (MDs) and nurses) and COVID-affected professionals (teachers, psychologists, and dentists). SQ and AQ were worse in adults, females, and high-education subjects. Feeling worse, having economic problems, depression, anxiety, irritability, and a high Calamity Experience Check List (CECL) score during COVID were significantly associated with poor SQ and AQ. Shorter sleep duration, increased latency, poor nutrition, low physical activity, increased mobile and social network use, more negative and less positive attitudes and behaviors were associated with poor AQ. Conclusions: The SQ logistic regression showed gender, morbidities, CECL, and awakenings as relevant, whereas, for AQ, relevant variables further included age and physical activity. Aiming to have a high stress compliance, each individual should sleep well, have important control of their mood, practice positive behaviors while dismissing negative behaviors and attitudes, practice exercise, have adequate nutrition, and beware of technologies and dependences.

## 1. Introduction

Sleep is critical for mental physical health and for survival. Sleep becomes particularly relevant when facing stressing situations such as those faced during the COVID-19 pandemic [1,2].

Sleep quality (SQ) is considered a basic sleep variable impacting the individual’s daily wellbeing. It may be considered a subjective evaluation item of sleep satisfaction or may be objectively quantified by polysomnography and actigraphy. The subjective dimension is quantifiable by several instruments, among which the Pittsburg Sleep Quality Index (PSQI) emerges as the international standard [3]. Nevertheless, the Jenkins Sleep Scale [4], the Medical Outcomes Study-Sleep Scale (MOS-SS) [5], the Insomnia Severity Scale [6], the Visual Analogue [7], or the Likert Scale are also valid instruments that assess subjective SQ.

The Self Rating Questionnaire for Sleep Quality evaluates SQ, AQ, and somatic complaints in 20 questions with four levels per question, which altogether provide a single score [8]. This scale has been used in several polysomnographic studies, but the correlations with objective sleep parameters are moderate, and no distinction exists between sleep and awakening quality [9].

The objective sleep continuity variables (sleep latency, number of awakenings <5 min, wake after sleep onset (WASO), and sleep efficiency) are good indicators of SQ 10]. The National Sleep Foundation addresses exact value ranges associated with good or poor SQ, with some of them being used for all ages, while others vary according to age [10].

During the COVID-19 pandemic, SQ was particularly affected. A study involving Greece, Switzerland, Austria, Germany, France, and Brazil reported worse SQ in 31.3% of participants, and 15% of them characterized their sleep as bad [11]. Poor SQ prevalence varied between 18.4% in China [12] and much higher numbers in Italy, 55.3% [13] and 57.1% [14]. However, 75.2% of Chinese adults in home isolation for more than 77 days rated their SQ as very good [15]. This might suggest that some cultural and/or regional differences might also affect subjective SQ [16,17].

The factors associated with poor sleep quality are anxiety, stress, or depression [10,11,13,14,15,16,17,18,19,20,21,22,23,24], marked changes in sleep/wake rhythm [10,18,25], insomnia [18], not exercising [19,23], increased use of multimedia such as television (TV) viewing and high personal computer (PC) use [23], negative attitude toward COVID-19 control measures [26], higher education level [19,26], family burden [24], low social capital [13,19], reduced flexibility [27], negative mood and “shielding” from the virus [24,28], COVID-19-related worries, or decreased resilience [29].

Protective factors are high social capital [13], social support, staying busy, and using free time in relaxing activities [30]. Furthermore, SQ mediates the relationship between physical activity and quality of life (QoL) [12].

The more affected groups were healthcare workers, mostly those working in frontline services [7,14,24,31,32,33], young people [11], older age [7], females [10,11,18,19], males performing highly demanding jobs [34], people suffering from death in the family [20], or those living in places with high COVID-19 prevalence [14].

Despite the predominantly negative impact upon SQ, some data suggest that the lockdown effect is bimodal since this factor improves in a subset of individuals [35,36]. This occurred in subsets of adults [12], particularly insomniacs. Morning awakening is a critical functional period due to the association with sleep inertia, sleep restoration, increased blood pressure and cortisol circadian rhythms, increased risk of morning headaches [37], awakening epilepsies, cardio- or cerebrovascular acute disorders, and sleep-awakening symptoms. Poor quality of awakening is often associated with excessive daytime sleepiness [38], insomnia symptoms, sleep/wake phase delay syndrome [39], sleep deprivation/insufficient sleep, and nonrestorative sleep [40]. In spite of being a key factor for individuals’ daily wellbeing, the quality of morning awakening is not currently investigated in population surveys.

The aim of this study was to characterize the impact of the COVID-19 pandemic on sleep and awakening qualities of a large, diversified population. A further objective was to identify significant associations and risks in terms of demography, health, other sleep variables, confinement mood, attitudes, behaviors, mood scales (depression, anxiety, irritability, and worries), economic problems, and health-related/risk behaviors (physical activity, multimedia use, nutrition, toxic habits, and addictions, i.e., habits, practices, activities, or personal attributes that either enhance or put at risk the overall health).

## 2. Materials and Methods

The Survey Legend^®^ platform was used. Surveys were anonymous, for adults (>18 years), allowing data analysis and statistical use. The first page included purpose, authors, ethical reference, contact person, and supporting entities. It was online during the first COVID-19 wave, from April to August 2020. Although the surveys were adapted to each main subgroup, they all had a core and common structure used in this study.

The overall project was approved by CENC’s Ethical Committee 1/2020. There was no funding, public or private, and no conflict of interests.

Online surveys were used, and 5479 individuals were included and distributed as follows: the general population (GP) (*N* = 972; age = 49.2 ± 13.5; 77.1% females); sleep disorder patients (SDPs) (*N* = 1261; age = 57.8 ± 14.2; 42.2% females); professionals COVID-involved: medical doctors and nurses (*N* = 2794; age = 44.8 ± 13.4; 72.7% females); professionals COVID-affected: teachers, psychologists, and dentists (*N* = 452; age = 45.0 ± 8.9; 88.1% females). Age and gender differences of the subgroups are considered an important issue, since they achieve population diversity. The study includes data obtained in the Portuguese mainland and islands (Madeira and Azores).

Surveys addressed the following topics: demographics, health status; confinement mood, attitudes, and behaviors; Calamity Experience Check List (CECL); sleep; physical activity (PA); multimedia use; nutrition; toxic habits and addictions.

Demographics, in addition to conventional information, included the number of people living together during the pandemic.

Health status included yes/no questions to the following topics: being healthy (subjective) or suffering from sleep, psychiatric, neurologic, cardiovascular, respiratory, allergies, gastrointestinal, rheumatologic, endocrinologic/metabolic, autoimmune, orthopedic, cancer, renal, dermatologic, hematologic, gynecologic, urologic, ear/nose/throat (ENT), or ophthalmologic disorders, chronic pain, fatigue, and dizziness.

The morbidities index (MI) is the sum of all referred morbidities at baseline with respect to COVID-19 in terms of worsening (morbidities worsening index (MWI)) and improvement (morbidities improvement index (MII)).

Confinement attitudes and behaviors were evaluated by yes/no answers. The average and number of both positive and negative attitudes and behaviors were computed per subject and used in this study. Positive attitudes were the following: felt OK; had less stress; made important discoveries. Negative attitudes were the following: fed up or tired; cannot stand it; loneliness; missing family or friends; felt in imprisonment/claustrophobia: had worries and fears; had unexpected conflicts; cannot stand companion; cannot stand children; cannot stand elderly; fed up with the children’s tele-school. Positive behaviors were the following: tidying up; new type of work: phone friends; decided life changes; wrote a book, articles, or memories; learned new abilities; gardening/agriculture; invented funny or spiritual things; worked; walking/gym/sports; reading/music/studying; domestic work. Negative behaviors were the following: developed new addictions; get bored; mourned all time; slept as much as possible.

Calamity mood data were obtained using 1–10 VASs (visual analogue scales). The Calamity Experience Check List (CECL) was computed by averaging, for each subject, the scales of depression, anxiety, irritability, and worries (Tomé et al. in publication). The reason for such a selection is related to the fact that these symptoms, in clinical practice, are quite frequently associated with sleep complaints.

Sleep data included the following data relative to weekdays and weekends during COVID-19: sleep schedules, subjective sleep duration in hours, sleep latency in minutes, number of awakenings, and sleep quality (SQ) and awakening quality (AQ) obtained using a 1–10 VAS.

Physical activity was quantified in terms of intensity (null, mild, moderate, and intense) and frequency (hours/week).

Screen time (television, social networks, mobile phone, and gaming use) was quantified in hours/day.

Nutrition included daily meals, and scores for the recommended intake frequencies were calculated [41] (Appendix A).

Toxic habits—Smoking and alcoholic consumption were categorized as yes/no, and daily cigarettes and glasses of beer, wine, aperitive wine, and brandy were assessed, respectively. Drug use was quantified as no, occasionally, sometimes, or regularly

Statistics—Qualitative variables were described by absolute and relative frequencies, while quantitative variables were calculated as the mean or median depending on data distribution. Normality was tested by the Kolmogorov–Smirnov test. Most continuous variables had a normal distribution except for CECL, get up time for weekdays and weekends, and cigarettes per/day. The 25th percentile (P25) values for SQ and AQ were calculated for the pre-COVID-19 period, and the variables were dichotomized as P25 or below (poor quality) and >P25 (good quality). The effect of the variables of interest was evaluated according to the variables in question using a paired *t*-test to compare SQ and AQ before (routinely occurring prior to the pandemic) and during COVID-19, chi-square tests (qualitative answers/frequency tables), and ANOVA (unidirectional analysis of variance) with post hoc Bonferroni tests. A logistic regression model was performed for SQ and AQ with age, sex, CECL, number of morbidities, sleep latency, sleep awakenings on weekdays during COVID-19, and physical activity intensity as covariates. All tests were performed using SPSS^®^v25. Statistical significance was set at 0.05.

## 3. Results

### 3.1. SQ and AQ before and during COVID-19

Data from SQ and AQ before and during COVID-19 were compared; they are shown in Table 1.

The reduction in SQ and AQ during COVID-19 was significant: *t* = 32,456, *n* = 4222, *p* < 0.001 and *t* = 25.06, *n* = 4213, *p* < 0.001, respectively. Before COVID-19, 21.9% had poor SQ; this number increased to 36.3% during COVID-19; the values for AQ were 22.8% before and 34.6% during the COVID-19 period. However, comparing before and during COVID-19 periods, 4.2% and 4.8% of the individuals improved their SQ and AQ, and 44.6% and 48.8% changed neither of them.

### 3.2. SQ and AQ during COVID-19

#### 3.2.1. Demographic Data

Differences considering demographic data are shown in Table 2. SQ and AQ were worse for adults (30–64 years), females, and subjects with higher education. Differences concerning civil status were only significant for AQ, which were better for married people (χ^2^ = 19.767; *p* = 0.001).

#### 3.2.2. Mood and Problems

The differences between the two groups (poor and good SQ and AQ) were significant for mood and economic problems.

During the 2020 confinement, lower values (feeling worse) were significant for both poor SQ and poor AQ; in terms of the economic problems, depression, anxiety, irritability, and the CECL, these were increased (i.e., more problems/more depressed, etc.) for poor SQ and AQ (Table 3).

#### 3.2.3. Sleep

Fewer differences were observed in sleep variables for the comparisons between poor and good SQ and AQ (Table 4).

For SQ, only the get up time on weekdays (later for poor SQ) and the higher values of sleep latency and awakenings, both on weekdays and weekends, were significantly different. 

For AQ, differences concerned mostly weekdays in relation to earlier get up time, shorter sleep duration, and longer sleep latency.

#### 3.2.4. Health-Related and Risks Behaviors

Health-related behaviors (HRBs) are presented in Table 5. For SQ data, these results were ambiguous since participants with poor SQ ate more recommended and less non-recommended food, smoked less, and had lower levels of TV and game dependence.

However, for AQ, the data are quite clear, i.e., participants with poor AQ practiced less PA, while they ate fewer meals and less recommended food and more non-recommended food, with a poor yes/no proportion; they had higher use of social networks and mobile phones, as well as a higher dependence of social networks, but they consumed fewer alcoholic drinks.

#### 3.2.5. Attitudes and Behaviors

The associations between attitudes and SQ/AQ are detailed in Table 6.

Poor SQ was associated with negative attitudes and behaviors, while AQ was significantly lower in terms of negative attitudes and higher in terms of negative behaviors and positive attitudes and behaviors, as well as in negative behaviors, which was unexpected.

#### 3.2.6. Health Prior to and during COVID-19

A small number of subjects (*n* = 122) were COVID-19-infected, of which 30 were asymptomatic, with poor SQ and AQ in eight and seven respondents, respectively; 81 had mild symptoms, with poor SQ and AQ in 46 and 37 respondents, respectively; seven had pneumonia, with poor SQ and AQ in four and four respondents, respectively; four were hospitalized, none of which had poor SQ or AQ. For SQ, the chi-square value was 17.561 (*p* < 0.002) and, for AQ, the chi-square value was 8.921 (*p* = 0.063).

The statistical links between SQ/AQ and body mass index (BMI)/health are presented in Table 7.

SQ was poorer with an increased number of morbidities, a higher level of worsening morbidities, and a higher level of improving morbidities, whereas poor AQ was associated with a lower BMI, a higher level of worsening morbidities, and a lower level of improving morbidities.

### 3.3. Logistic Regression

A logistic regression model for SQ and AQ was performed for variables significantly associated with poor SQ and poor AQ.

#### 3.3.1. Poor Sleep Quality

Male gender (odds ratio (OR) = 0.725; 95% confidence interval (CI): 0.596–0.882; *p* = 0.001) was the only significant protective factor for poor SQ. A higher number of morbidities (OR = 1.184; 95% CI: 1.121–1.252; *p* < 0.001) and a higher average CECL (OR = 1.441; 95% CI: 1.373–1.512; *p* < 0.001) were risk factors. Regarding sleep variables, a high number of awakenings on weekdays during COVID-19 was the only variable representing a significantly increased risk factor for poor SQ (Table 8). 

#### 3.3.2. Poor Awakening Quality

Being older (OR = 0.993; 95% CI: 0.986–0.999; *p* = 0.033), male gender (OR = 0.762; 95% CI: 0.626–0.926; *p* = 0.006), and performing light (OR = 0.624; 95% CI: 0.406–0.961; *p* = 0.032) or moderate (OR = 0.612; 95% CI: 0.402–0.943; *p* = 0.026) physical activities were significant protective factors for poor AQ. A higher number of morbidities (OR = 1.168; 95% CI: 1.107–1.233; *p* < 0.001) and a higher average calamity scale score (OR = 1.398; 95% CI: 1.333–1.467; *p* < 0.001) were also risk factors for poor AQ (Table 9).

## 4. Discussion

This study brings new insights to the sleep field concerning both self-reported sleep quality and final awakening quality (SQ and AQ) in a large population study recruited from different regions of Portugal (mainland and islands) and from different subgroups, including sleep disorder patients.

The discussion involves the following aspects: (1) the used survey; (2) the SQ evaluation in the survey; (3) the use of AQ; (4) the prevalence values of SQ and AQ; (5) the associated factors and their functional meaning; (6) the risks of poor SQ and AQ.

The survey was constructed considering the following objectives: determination of individual and environmental factors influencing pandemic compliance, health, and sleep quality. Questions focused on standard demographic data, according to previous team surveys [42] or World Health Organization (WHO) and European Union (EU) question surveys or recommendations, as well as on clinical experience regarding schedules, habits, and attitudes. The existence of open questions allowed an enlarged scope in some respects. Most variables were either quantitative or dichotomic (yes/no). Sleep was only a chapter, whereby health evaluated both sleep and medical disorders. This differed from the International COVID-19 Sleep Study (ICOSS) survey [43], which focused on sleep/sleep disorders and used standard questionnaires; another major difference was the size: 50 questions for the ICOSS and 177 in our case. The ICOSS survey was international, while the present survey was focused on Portugal.

SQ evaluation is, in most surveys, done by using standard questionnaires, such as the PSQI [3], while AQ evaluation uses the SSA (Scale for Sleep and Awakening quality) questionnaire [8,9]. The different approach used by our group, via the application of VASs, was primarily to do with the extensive and multifocal questionnaire used and the need to reduce the number of questions as much as possible. Moreover, the questions relative to sleep schedules, sleep duration, and latencies implied both weekdays and weekends, prior to and during COVID-19. VASs achieve high response rates and high levels of completion, and there is significant evidence demonstrating their reliability in terms of inter-rater reliability and test–retest reliability; the differences observed between SQ and AQ favored the inexistence of response spreading, which is possible when using VASs [44]. A lack of precision, subjectiveness, and the frequent pessimistic attitude in evaluating complaints are common, especially in sleep disorder patients [45]. Altogether, these arguments led to a simple VAS with clear limits.

The use of an AQ scale reflects the recognition of the importance of the transition from sleep to awake, as discussed previously [8,9], together with its possible relevance for daytime consequences in terms of mood and behavior. Effectively, the AQ evaluation provided more frequent and consistent results than the SQ scale itself. This was shown where physical activity had a protective effect for poor AQ but not for SQ after adjusting for other variables such as age and gender. These results differ from those reported in other studies, where physical exercise practice is often associated with SQ improvement [46]. This relationship with AQ should be further studied since the AQ is also often dependent on SQ. In a recent study, the relationship between SQ of the previous night of sleep and a better cognitive performance (i.e., AQ performance) on the subsequent day was clearly shown for a sample of young adults [47].

The prevalence values of poor SQ during COVID-19, albeit high (SQ = 36.3% and AQ 34.6%), are lower than those obtained by Italian authors [13,14] but higher than those observed in China [15]; these differences are likely due to the differences in COVID-19 severity between countries and its consequent impact upon mood and perception [48]. Compared with the pre-COVID-19 period, the average values of SQ and AQ were in fact significantly lower. However, it must be stated that, for SQ, 4.2% of respondents improved and 44.6% had no change; the corresponding values for AQ were 4.8% and 48.8%, respectively. This means that deterioration was achieved by half of the population, whereas the other half remained equal or better; this shows a high resilience to stressing situations, a fact already reported in a recent study for a large population under the COVID-19 pandemic [49].

Our sample had a wide age distribution, and it is currently assumed that older age negatively impacts SQ. In this study, older people reported a significantly better AQ, while age differences for SQ were not found. This relationship with aging might be explained by the appearance of feeling refreshed after a night of sleep and easiness of awakening. In line with these results in the elderly, the wake drive predominates in relation to the weakening of the sleep drive [50].

In our study, women reported a significantly worse SQ and AQ than men. Poorer SQ in females was observed by others [51,52]; this is likely due to the specificities of sleep in the female gender [53]. Women tend to have a higher prevalence of poor sleep, independent of depression, sociodemographic, and lifestyle factors [51], and the difference is most likely a biological issue [53].

Both SQ and AQ are significantly associated with several factors. Like other studies, we found poor SQ and AQ in young adults [11], females [10,11,18,19], higher-education individuals [19,26], and healthcare workers (medical doctors (MDs) and nurses) [7,14,24,31,32,33]; however, civil status was not an influencing factor, except for AQ, and the oldest group had higher SQ and AQ levels. The differences in married people are in line with the data concerning the number of people living together; SQ was better for two persons while AQ was better for 1–4 persons. Both SQ and AQ were poorer in larger households. Better results in the elderly might be related to a better acceptance of living circumstances, but we have no data to prove this.

Mood evaluation, the calamity scale, and facing economic problems were significantly associated with SQ and AQ. The data concerning anxiety, stress, depression [10,11,13,14,15,16,17,18,19,20,21,22,23,24], and worries [29] are in line with published results for SQ; the differences obtained concern the calamity scale and economic concerns. This might be due to the time of collection, since, in the present study, although undertaken during the first wave of COVID-19, data were collected until August, with people already facing 5 months of COVID-restrictions and direct and indirect economic impacts with, in many cases, job losses. The cited studies were performed at an earlier time point of the pandemic situation. This study highlights once more the relevance of SQ in mood and behavior especially for stressful situations [1,2].

The associations between SQ and other sleep variables were restricted to sleep latency and night waking on weekends and weekdays, as also established by others [10]. There was no association between SQ and sleep duration as conventionally assumed, although it existed for AQ. Both SQ and AQ were associated with get up time on weekdays, but in opposite directions. These discrepancies enhance the difficulties in the evaluation of subjective sleep data, reinforcing the need for objective data.

HRBs were significantly associated with AQ, specifically, physical activity [19,23,54], while reinforcing the need for proper nutrition in terms of meals and food quality [53,55]. This enhances the need to moderate the use of social networks and mobile phones, and it highlights the dependence problem of modern technologies [23]. The timings of both exercise [56] and nutrition [57], although relevant for sleep, were not evaluated, due to the need to restrict the number of survey questions.

Attitudes and behaviors during COVID-19 were evaluated in detail in this study. Negative attitudes were associated with poor SQ [24,26,28] and AQ. Positive attitudes and behaviors were associated with good AQ; however, surprisingly, this also occurred for negative behaviors. The knowledge, attitudes, and practices that people hold toward the disease play an integral role in determining a society’s readiness to accept behavioral change measures from health authorities in response to the COVID-19 pandemic, according to the study by Azlan et al. (2020) [58]. Data presented by Li et al. (2020) [59] corroborate these results, which showed that participants’ knowledge about COVID-19 was positively related to social participation and precautionary behavior. However, when there was greater perceived severity of the situation, together with the perceived controllability as negative, there was an association with sleep problems and negative emotions.

COVID19 infection, in a relatively small group, with many asymptomatic subjects, was significantly associated with poor SQ; the time delay between infection and the survey answer was, however, not collected. Available data show worse SQ in more severe patients [60]. 

Underweight was associated with poor AQ. Increased morbidities and a higher level of worsening morbidities were associated with poor SQ and AQ, even after adjusting for age and gender through the logistic model. However, improving morbidities had a clear-cut association with good AQ, whereas the effect was surprisingly opposite with SQ. This apparently odd result might reflect the difficult evaluation of subjective SQ, which, linked to multiple components of sleep, such as latency, duration, and interruptions, is liable to subjective misperception [45,61].

The self-reported cross-sectional nature of this report is a limitation that needs to be addressed. This study was not a national prevalence study and was based online, which can be considered a selection bias. Although the study did not include a national randomized sample, we consider it to be a representative study, since we had answers from all Portuguese regions, including the islands (Madeira and Azores). An aspect that also brought some strength to our study was the collection of many relevant variables that may impact daily behaviors, as well as affect the response to social restrictions, such as general health, marital status, children, and socioeconomic status.

## 5. Conclusions

This study provides a global view of what is recommended during COVID-19 or equivalent catastrophic periods. To sleep well, while having a high stress compliance, each individual should not only sleep but also have important control of their mood, practice positive behaviors, dismiss negative behaviors and attitudes, practice physical exercise, take care with meals and adequate nutrition, and beware of technologies and dependences.

## Figures and Tables

**Table 1 ijerph-18-03506-t001:** Data from sleep quality (SQ) and awakening quality (AQ) before and during COVID-19.

Variable	Value	SQ Pre COVID	SQCOVID	AQ Pre COVID	AQ COVID
Number	Valid	4230	4232	4227	4223
	Missing	1249	1247	1252	1256
Median		7	6	7	6
Minimum		1	1	1	1
Maximum		10	10	10	10
Percentiles	25	5	4	5	4
	50	7	6	7	6
	75	8	7	8	7
	90	9	8	9	8

**Table 2 ijerph-18-03506-t002:** Associations of sleep and awakening quality with demographic data.

**Age Groups**		**Young**	**Adults**	**Elderly**	**Total (%)**	***p*** **-Value**				
Sleep Quality P25	Poor	166 (3.0)	1544 (28.3)	269 (4.9)	1979 (36.3)	<0.001				
	Good	317 (5.8)	2516 (46.1)	645 (11.8)	3478 (63.7)					
Awakening Quality P25	Poor	165 (3.0)	1499 (27.5)	224 (4.1)	1888 (34.6)	<0.001				
	Good	318 (5.8)	2561 (47.0)	690 (12.6)	3569 (65.4)					
Total (%)		483 (8.8)	4060 (74.4)	914 (16.8)	5457 (100.0)					
**Gender**		**Male**	**Female**	**Total** **(%)**	***p*** **-Value**					
Sleep Quality P25	Poor	505 (9.2)	1482 (27.1)	1987 (36.3)	<0.001				
	Good	1264 (23.1)	2228 (40.6)	3492 (63.7)						
Awakening Quality P25	Poor	486 (8.9)	1408 (25.7)	1894 (34.6)	<0.001					
	Good	1283(23.4)	2302 (42.0)	3585 (65.4)						
Total (%)		1769 (32.3)	3710 (67.7)	5479 (100.0)						
**Education**		**Low**	**High**	**Total (%)**	***p*** **-Value**					
Sleep Quality P25	Poor	268 (4.9)	1719 (31.4)	1987 (36.3)	<0.001					
	Good	624 (11.4)	2868 (52.3)	3492 (63.7)						
Awakening Quality P25	Poor	232 (4.2)	1662 (30.4)	1894(34.6)	<0.001					
	Good	660 (12.1)	2925 (53.3)	3585 (65.4)						
Total (%)		892 (16.3)	4587 (83.7)	5479 (100.0)						
**Civil status**		**Married**	**Single**	**Widow**	**Divorced**	**Union**	**Total (%)**	***p*** **-Value**		
Sleep Quality P25	Poor	1016 (18.6)	458 (8.4)	50 (0.9)	206 (3.8)	255 (4.6)	1987 (36.3)	0.316		
	Good	1876 (34.2)	790 (14.4)	85 (1.6)	315 (5.7)	422 (7.8)	3492 (63.7)			
Awakening Quality P25	Poor	928 (17.0)	472 (8.6)	41 (0.7)	196 (3.6)	255 (4.7)	1894 (34.6)	<0.001		
	Good	1964 (35.8)	776 (14.2)	94 (1.7)	325 (5.9)	422 (7.7)	3585 (65.4)			
Total (%)		2892 (52.8)	1248 (22.8)	135 (2.5)	521 (9.5)	677 (12.4)	5479 (100.0)			
**People at the** **Household**		**1**		**2**	**3**	**4**	**5**	**≥6**	**Total (%)**	***p*** **-Value**
Sleep Quality P25	Poor	324 (6.3)		645 (12.5)	436 (8.5)	385 (7.5)	108 (2.1)	76 (1.5)	1974 (38.3)	0.029
	Good	439 (8.5)		1173 (22.8)	685 (13.3)	605 (11.7)	167 (3.2)	113 (2.2)	3182 (61.7)	
Awakening Quality P25	Poor	322 (6.2)		631 (12.2)	408 (6.9)	358 (6.9)	255 (4.7)	65 (1.3)	1886 (36.6)	0.018
	Good	441 (13.5)		1187 (23.0)	632 (12.3)	632 (12.3)	422 (7.7)	124 (2.4)	3270 (63.4)	
Total (%)		763 (14.8)		1818 (35.3)	1121 (21.7)	990 (19.2)	275 (12.4)	189 (3.7.0)	5156 (100%)	

Pearson’s chi-square test for all group comparisons; values are given as the frequency (percentage); P25-represents the 25th percentile of the score distribution; young-≤29 years; adults-30–64 years; elderly- ≥65years.

**Table 3 ijerph-18-03506-t003:** Differences in sleep and awakening quality relative to feelings and mood during COVID-19.

**Sleep Quality P25**		***N***	**Mean**	**SE**	**Minimum**	**Maximum**	***Z***	***p*-Value**
How are you living during confinement	Poor	1168	6.31	6.21	1	10	33.025	<0.001
	Good	3774	6.66	6.60	1	10		
How are your economic problems?	Poor	1186	3.38	3.25	1	10	51.831	<0.001
	Good	3790	2.87	2.80	1	10		
How is your depression?	Poor	1190	4.17	4.03	1	10	42.987	<0.001
	Good	3819	3.65	3.58	1	10		
How is your anxiety?	Poor	1190	5.13	4.99	1	10	42.71	<0.001
	Good	3804	4.58	4.50	1	10		
How is your irritability?	Poor	1186	4.95	4.81	1	10	18.996	<0.001
	Good	3817	4.58	4.50	1	10		
How are your worries vs. uncertainty?	Poor	1185	6.37	6.23	1	10	25.407	<0.001
	Good	3788	5.96	5.88	1	10		
Calamity Experience Check List	Poor	1197	5.11	5.00	1	10	47.941	<0.001
	Good	3847	4.64	4.58	0.75	10		
**Awakening Quality P25**		***N***	**Mean**	**SE**	**Minimum**	**Maximum**	***Z***	***p*-Value**
How are you living during confinement?	Poor	1845	6.03	1.784	1	10	290.166	<0.001
	Good	3097	6.91	1.741	1	10		
How are your economic problems?	Poor	1878	3.33	2.237	1	10	74.496	<0.001
	Good	3098	2.79	2.06	1	10		
How is your depression?	Poor	1879	4.58	2.353	1	10	364.663	<0.001
	Good	3130	3.3	2.266	1	10		
How is your anxiety?	Poor	1883	5.6	2.39	1	10	408.062	<0.001
	Good	3111	4.17	2.439	1	10		
How is your irritability?	Poor	1879	5.51	2.42	1	10	359.52	<0.001
	Good	3124	4.16	2.443	1	10		
How are your worries vs. uncertainty?	Poor	1878	6.74	2.236	1	10	254.552	<0.001
	Good	3095	5.64	2.457	1	10		
Calamity Experience Check List	Poor	1892	5.57	1.92	1	10	530.367	<0.001
	Good	3152	4.26	1.98	0.75	10		

P25—represents the 25th percentile of the score distribution; SE—standard error; *Z*-scores indicate the differences between the groups.

**Table 4 ijerph-18-03506-t004:** Differences in sleep variables for the comparisons between poor and good SQ and AQ during COVID.

**Sleep Quality P25**		***N***	**Mean**	**SE**	**Minimum**	**Maximum**	***Z***	***p*-Value**
Get up time weekdays (hours)	Poor	1007	8.05	1.535	3.75	19.25	15.825	<0.001
	Good	3449	7.85	1.428	3.00	19.00		
Get up time weekends (hours)	Poor	1012	9.00	1.511	5.00	15.00	3.732	0.053
	Good	3461	8.90	1.506	3.25	19.00		
Time into bed weekdays (hours)	Poor	980	0.00	1.548	18.00	8.00	2.863	0.091
	Good	3342	23.51	1.516	18.00	15.00		
Time into bed weekends (hours)	Poor	986	0.35	1.659	18.00	11.50	3.716	0.054
	Good	3330	0.24	1.603	18.00	12.00		
Sleep duration weekdays (hours)	Poor	980	6.69	1.615	00.25	16.00	0.042	0.838
	Good	3302	6.68	1.641	00.13	16.00		
Sleep duration weekends (hours)	Poor	975	7.41	1.849	00.34	16.00	0.881	0.348
	Good	3295	7.48	1.973	00.13	19.00		
Sleep latency weekdays (min)	Poor	928	34.16	35.017	0	300	3.937	0.047
	Good	3153	31.68	32.972	0	240		
Sleep latency weekends (min)	Poor	927	34.06	35.841	0	300	5.6	0.018
	Good	3140	31.04	33.693	0	302		
Night awakenings weekdays (number)	Poor	802	3.00	3.128	1	30	7.158	0.007
	Good	2524	2.72	2.451	0.5	30		
Night awakenings weekends (number)	Poor	760	2.59	2.260	1	30	9.994	0.002
	Good	2420	2.34	1.689	0.1	30		
**Awakening Quality P25**		***N***	**Mean**	**SE**	**Minimum**	**Maximum**	***Z***	***p*-Value**
Get up time weekdays (hours)	Poor	1589	7.83	1.47	3.00	19.25	4.799	0.029
	Good	2867	7.93	1.45	3.00	17.00		
Get up time weekends (hours)	Poor	1592	8.93	1.51	4.00	19.00	0.069	0.794
	Good	2881	8.92	1.51	3.25	18.00		
Time into bed weekdays (hours)	Poor	1545	23.51	1.51	18.00	09.50	0.426	0.514
	Good	2777	23.54	1.53	18.00	15.00		
Time into bed weekends (hours)	Poor	1551	0.28	1.65	18.00	12.00	0.141	0.708
	Good	2765	0.26	1.60	18.00	12.00		
Sleep duration weekdays (hours)	Poor	1540	6.57	1.58	0.29	16.00	12.005	0.001
	Good	2742	6.75	1.66	0.13	16.00		
Sleep duration weekends (hours)	Poor	1539	7.40	1.87	0.29	16.00	2.85	0.091
	Good	2731	7.50	1.99	0.13	19.00		
Sleep latency weekdays (min)	Poor	1478	33.82	34.90	0.00	300.00	5.165	0.023
	Good	2603	31.35	32.59	0.00	240.00		
Sleep latency weekends (min)	Poor	1475	33.09	34.97	0.00	300.00	3.68	0.055
	Good	2592	30.95	33.76	0.00	302.00		
Night awakenings weekdays (number)	Poor	1228	2.85	2.70	1.00	30.00	0.989	0.32
	Good	2098	2.75	2.59	0.50	30.00		
Night awakenings weekends (number)	Poor	1163	2.43	1.81	0.10	30.00	0.414	0.52
	Good	2017	2.39	1.86	0.50	30.00		

P25—represents the 25th percentile of the sample distribution; SE—standard error; hours are given in decimal values for local time and in 24 h format; *Z*-scores indicate the differences between the groups.

**Table 5 ijerph-18-03506-t005:** Differences in sleep quality and awakening quality relative to health-related and risk behaviors.

**SLEEP Quality P25**		**N**	**Mean**	**SE**	**Minimum**	**Maximum**	**Z**	***p*-Value**
Physical activity during COVID (hours)	Poor	905	2.66	4.027	0	48	0.518	0.472
	Good	2229	2.77	3.828	0	60		
Meals/day during COVID	Poor	1162	3.81	0.901	1	5	1.556	0.212
	Good	2942	3.85	0.906	1	5		
Food Recommended	Poor	1161	1.3167	0.14009	1	2	2.144	0.143
	Good	2940	1.31	0.13375	1	2		
Food Recommended YES	Poor	1147	5.37	2.324	0	14	4.551	0.033
	Good	2917	5.26	2.211	1	14		
Food Recommended NO	Poor	1154	11.45	2.452	1	17	4.133	0.042
	Good	2932	11.63	2.324	1	17		
Food Recommended YES/NO proportion	Poor	1146	0.5421	0.39774	0	4.67	2.144	0.143
	Good	2915	0.5158	0.36136	0.06	4.67		
Alcohol during COVID	Poor	755	9.1836	26.8108	0	563.5	0.052	0.819
	Good	1866	9.3843	17.0966	0	139.5		
Number cigarettes/month during COVID	Poor	172	11.23	8.907	0	60	4.597	0.032
	Good	376	12.94	8.563	0	40		
TV/Day during COVID (hours)	Poor	1055	3.199	2.4342	0.1	20	3.044	0.081
	Good	2687	3.044	2.4365	0.1	20		
Social Networks during COVID (hours)	Poor	927	2.541	2.4236	0.1	20	2.539	0.111
	Good	2347	2.398	2.2852	0	20		
Mobile use/Day during COVID (hours)	Poor	1008	2.649	2.8126	0	20	1.485	0.223
	Good	2624	2.532	2.5031	0.1	20		
Games/Day during COVID (hours)	Poor	227	1.825	1.4432	0.1	14	0.526	0.469
	Good	672	1.925	1.9114	0.1	20		
TV dependence	Poor	1097	3.19	1.962	1	10	6.236	0.013
	Good	2812	3.38	2.122	1	10		
Social Networks dependence	Poor	1098	3.55	2.312	1	10	2.659	0.103
	Good	2817	3.68	2.4	1	10		
Games dependence	Poor	1090	1.59	1.392	1	10	4.063	0.044
	Good	2813	1.7	1.579	1	10		
Alcohol dependence	Poor	1094	1.45	1.171	1	10	0.308	0.579
	Good	2811	1.47	1.159	1	10		
**AWAKENING Quality P25**		**N**	**Mean**	**SE**	**Minimum**	**Maximum**	**Z**	***p*-Value**
Physical activity during COVID (hours)	Poor	1302	2.46	3.848	0	48	11.8	<0.001
	Good	1832	2.94	3.903	0	60		
Meals/day during COVID	Poor	1835	3.8	0.937	1	5	4.635	0.031
	Good	2269	3.87	0.877	1	5		
Food Recommended	Poor	1831	1.2997	0.13788	1	2	26.741	<0.001
	Good	2270	1.3217	0.13294	1	2		
Food Recommended (YES)	Poor	1810	5.09	2.266	0	14	26.402	<0.001
	Good	2254	5.45	2.213	1	14		
Food Recommended (NO)	Poor	1824	11.74	2.437	1	17	14.696	<0.001
	Good	2262	11.45	2.292	1	17		
Food Recommended YES/NO proportion	Poor	1808	0.4987	0.37094	0	4.67	14.171	<0.001
	Good	2253	0.5429	0.372	0.06	4.67		
Alcohol consumption during COVID	Poor	1159	7.8347	22.94508	0	563.5	11.188	<0.001
	Good	1462	10.5091	17.99032	0	139.5		
Number cigarettes/month during COVID	Poor	277	12.92	8.736	0	40	1.983	0.16
	Good	271	11.88	8.648	0	60		
TV/Day during COVID (hours)	Poor	1645	3.009	2.4141	0.1	20	3.094	0.079
	Good	2097	3.15	2.4527	0.1	20		
Social Networks during COVID (hours)	Poor	1504	2.565	2.4988	0	20	8.266	0.004
	Good	1770	2.331	2.1628	0.1	20		
Mobile use/Day during COVID (hours)	Poor	1624	2.761	2.7572	0	20	17.003	<0.001
	Good	2008	2.405	2.441	0.1	20		
Games/Day during COVID (hours)	Poor	389	1.919	1.9467	0.1	20	0.08	0.777
	Good	510	1.885	1.6897	0.1	20		
TV dependence	Poor	1731	3.3	2.058	1	10	0,689	0.407
	Good	2178	3.35	2.097	1	10		
Social Networks dependence	Poor	1736	3.84	2.407	1	10	21.279	<0.001
	Good	2179	3.49	2.34	1	10		
Games dependence	Poor	1731	1.69	1.606	1	10	0.916	0.339
	Good	2172	1.65	1.466	1	10		
Alcohol dependence	Poor	1733	1.48	1.206	1	10	0.509	0.476
	Good	2172	1.46	1.126	1	10		

P25—represents the 25th percentile of the sample distribution; SE—standard error; hours are given in decimal values for local time and in 24 h format; *Z*-scores indicate the differences between the groups.

**Table 6 ijerph-18-03506-t006:** Associations between attitudes/behaviors and sleep quality/awakening quality.

**Sleep Quality P25**		***N***	**Mean**	**SE**	**Minimum**	**Maximum**	***Z***	***p*-Value**
Positive attitudes (number)	Poor	1198	0.59	0.766	0	3	2.714	0.1
	Good	4044	0.55	0.724	0	3		
Negative attitudes (number)	Poor	1198	1.21	1.093	0	7	6.164	0.013
	Good	4044	1.13	1.045	0	7		
Positive behaviors (number)	Poor	1198	1.97	1.519	0	7	0.63	0.427
	Good	4045	1.93	1.517	0	8		
Negative behaviors(number)	Poor	1198	0.65	0.849	0	4	5.611	0.018
	Good	4045	0.59	0.811	0	5		
**Awakening Quality P25**		***N***	**Mean**	**SE**	**Minimum**	**Maximum**	***Z***	***p*-Value**
Positive attitudes (number)	Poor	1890	0.42	0.655	0	3	102.011	<0.001
	Good	3352	0.63	0.765	0	3		
Negative attitudes (number)	Poor	1890	1.45	1.128	0	7	263.127	<0.001
	Good	3352	0.97	0.972	0	7		
Positive behaviors (number)	Poor	1890	1.83	1.491	0	7	15.236	<0.001
	Good	3353	2	1.528	0	8		
Negative behaviors(number)	Poor	1890	0.70	0.88	0	4	40.237	<0.001
	Good	3353	0.55	0.779	0	5		

P25—represents the 25th percentile of the sample distribution; SE—standard error; *Z*-scores indicate the differences between the groups.

**Table 7 ijerph-18-03506-t007:** Relationship between sleep/awakening quality and body mass index (BMI)/health during COVID-19.

**Sleep Quality P25**		***N***	**Mean**	**SE**	**Minimum**	**Maximum**	***Z***	***p*-Value**
BMI	Poor	1191	25.73	5.01	13.96	48.48	0.016	0.898
	Good	4259	25.76	5.03	15.32	68.4		
Number of morbidities	Poor	1198	2.44	2.22	0	16	309.428	<0.001
	Good	4201	1.43	1.598	0	13		
Morbidities (worse)	Poor	1127	2.1	2.077	0	12	62.212	<0.001
	Good	3287	1.6	1.729	0	13		
Morbidities (better)	Poor	1122	0.55	1.276	0	11	23.857	<0.001
	Good	3276	0.37	0.974	0	9		
**Awakening Quality P25**		***N***	**Mean**	**SE**	**Minimum**	**Maximum**	***Z***	***p*-Value**
BMI	Poor	1887	25.38	4.94	14.82	48.48	15.952	<0.001
	Good	3563	25.95	5.05	13.96	68.4		
Number of morbidities	Poor	1894	2.09	2.077	0	16	176.466	<0.001
	Good	3505	1.42	1.59	0	13		
Morbidities (worse)	Poor	1779	2.37	2.042	0	13	395.207	<0.001
	Good	2635	1.3	1.541	0	10		
Morbidities (better)	Poor	1776	0.36	0.927	0	9	10.088	0.002
	Good	2622	0.46	1.143	0	11		

P25—represents the first quartile of the sample distribution; BMI—body mass index; the number of morbidities is relative to the pre-COVID situation; morbidities worse and better are for the post-COVID situation; SE—standard error; *Z*-scores indicate the differences between the groups.

**Table 8 ijerph-18-03506-t008:** Logistic regression analysis for poor sleep quality.

Variables	β	SE β	Wald’s χ^2^	df	*p*-Value	Odds Ratio	95% CI	
							Lower	Upper
Age	0.001	0.003	0.145	1	0.704	1.001	0.995	1.008
Sex (male)	−0.322	0.1	10.287	1	0.001	0.725	0.596	0.882
Number morbidities	0.169	0.028	36.185	1	< 0.001	1.184	1.121	1.252
Calamity Experience Check List	0.365	0.025	218.207	1	< 0.001	1.441	1.373	1.512
Sleep latency on weekdays during COVID-19	0.001	0.001	0.276	1	0.599	1.001	0.998	1.003
Awakenings on weekdays during COVID-19	0.038	0.019	3.903	1	0.048	1.039	1	1.078
Physical activity intensity			2.295	3	0.514			
None	−0.267	0.229	1.37	1	0.242	0.765	0.489	1.198
Light	−0.321	0.222	2.096	1	0.148	0.725	0.469	1.12
Moderate	−0.314	0.219	2.061	1	0.151	0.73	0.475	1.122
Constant	−1.866	0.29	41.513	1	< 0.001	0.155		

Hosmer–Lemeshow test χ2 (8) = 6.535, *p* = 0.588. Abbreviations: CI, confidence interval; df, degrees of freedom.

**Table 9 ijerph-18-03506-t009:** Logistic regression analysis for poor awakening quality.

Variables	β	SE β	Wald’s χ^2^	df	*p*-Value	Odds Ratio	95% CI	
							Lower	Upper
Age	−0.007	0.003	4.527	1	0.033	0.993	0.986	0.999
Sex (male)	−0.272	0.1	7.431	1	0.006	0.762	0.626	0.926
Number morbidities	0.156	0.027	32.329	1	< 0.001	1.168	1.107	1.233
Calamity Experience Check List	0.335	0.024	190.787	1	< 0.001	1.398	1.333	1.467
Sleep latency on weekdays during COVID-19	0.001	0.001	0.342	1	0.559	1.001	0.998	1.003
Awakenings on weekdays during COVID-19	0.019	0.019	1.034	1	0.309	1.019	0.983	1.057
Physical activity intensity			5.958	3	0.114			
None	−0.353	0.227	2.425	1	0.119	0.703	0.451	1.096
Light	−0.471	0.22	4.573	1	0.032	0.624	0.406	0.961
Moderate	−0.485	0.217	4.981	1	0.026	0.616	0.402	0.943
Constant	−1.235	0.285	18.814	1	< 0.001	0.291		

Hosmer–Lemeshow test χ^2^ (8) = 7.165, *p* = 0.519. Abbreviations: CI, confidence interval; df, degrees of freedom.

## Data Availability

Data availability is controlled by the task leader, Prof Teresa Paiva; furthermore, the data will exclusively be used throughout at least 2021 by the “COVID, Health, Sleep, Habits” group, which includes, among others, the present authors.

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
