# Peer review of "Sleep and Awakening Quality during COVID-19 Confinement: Complexity and Relevance for Health and Behavior"

_ijerph, 2021, doi:10.3390/ijerph18073506_

Round 1

Reviewer 1 Report

The current study aims to describe both risk and protective factors associated with sleep (SQ) and awakening (AQ) quality during the first wave of the current COVID-19 pandemic in Portugal. Amongst other parameters in their questionnaire, the authors implemented a novel tool of “Calamity Experience Check List (CECL)” to assess how each subject experienced their “confinement”, presumably during the respective lockdown.

There are several issues with the study, spanning from the methodology, to some of the main points as explained in the “Results” and “Discussion” sections:

- As the main novelty of the current study, the CECL should be carefully explained in the method section. The citation of a publication in preparation is not adequate for the reader to assess the quality and precision of the method, with the exception of back-to-back papers in the same journal.

- Despite the authors claim that the awakening quality (AQ) scale is (line 289; “in a sense a novelty of this work”), they have no citation to previous work on the SSA-1 and -2 scale also  evaluating awakening quality (Saletu et al., 1987; Rosipal et al., 2013), or to how their AQ scale differs from that one, and to which points it is superior or inferior.

- Perhaps a minor detail: the authors claim that N=5479 subjects participated to the study, however the individual groups sum only up to 5460, with no mention of the remaining participants (e.g dropouts?).

- Physical activity (PA) has been linked to higher sleep quality by most of studies so far (Dolezal et al., 2017). It is true, however, that the type of PA can impact sleep quality either positively or negatively (Jung et al., 2020; 10.17241/smr.2020.00542). A more precise classification of activity, focused on the type rather than the intensity of exercise) might resolve this ambiguity. Another very important parameter is the timing (hours close to bedtime) of physical activity (Chen et al., 2020; DOI: 10.1111/ecc.13233), which could explain ambiguous results in the current study, but is not included in the questionnaire. Last but not least, only physical intensity, but not frequency, is discussed in the result section.

- The timing of food intake might explain better the ambiguous results of SQ ~ “recommended food” relationship, particularly in a period where schedules shift due to mandatory or suggested house confinement.

- Could the content and timing (hours close to bedtime) of TV and games might be more important that the actual time spent?

- A more detailed classification of co-inhabitance, e.g “alone”, “married”, “with children”, “single parent with children”, and “co-inhabiting with non-relative”, would be more informative than civil state, particularly during a mandatory house confinement and social isolation, as that during the lockdown.

Author Response

We thank your comments which clearly increase the paper quality.

In order to address your review we include an answer to each item as follows:

  • As the main novelty of the current study, the CECL should be carefully explained in the method section. The citation of a publication in preparation is not adequate for the reader to assess the quality and precision of the method, with the exception of back-to-back papers in the same journal.
  • Answer: The reference of Tome et al is in a publication process, it is not cited in the references.
  • Despite the authors claim that the awakening quality (AQ) scale is (line 289; “in a sense a novelty of this work”), they have no citation to previous work on the SSA-1 and -2 scale also  evaluating awakening quality (Saletu et al., 1987; Rosipal et al., 2013), or to how their AQ scale differs from that one, and to which points it is superior or inferior.
  • Answer: The references are included and comparisons discussed
  •  
  • Perhaps a minor detail: the authors claim that N=5479 subjects participated to the study, however the individual groups sum only up to 5460, with no mention of the remaining participants (e.g dropouts?).
  • Answer- The numbers were corrected, there are no dropouts.
  •  
  • Physical activity (PA) has been linked to higher sleep quality by most of studies so far (Dolezal et al., 2017). It is true, however, that the type of PA can impact sleep quality either positively or negatively (Jung et al., 2020; 10.17241/smr.2020.00542). A more precise classification of activity, focused on the type rather than the intensity of exercise) might resolve this ambiguity. Another very important parameter is the timing (hours close to bedtime) of physical activity (Chen et al., 2020; DOI: 10.1111/ecc.13233), which could explain ambiguous results in the current study, but is not included in the questionnaire. Last but not least, only physical intensity, but not frequency, is discussed in the result section.
  • Answer - The number of answers obtained for details in physical activity (type, location, individual or group) are much lower then the 2 questions used (intensity and duration) and therefore were discarded; furthermore the WHO reccommendations are hours per week. The timings of exercise  were not asked, but during lockdown and lockdown ease gymnasium and sportive facilities were closed, therefore most physical activity was “likely” performed during the day. We stress this dimension at the discussion. Reference is added
  •  
  • The timing of food intake might explain better the ambiguous results of SQ ~ “recommended food” relationship, particularly in a period where schedules shift due to mandatory or suggested house confinement.
  • Answer - The reference for recommended food is presented in the supplementary material. Basically, it provides the desirable intake frequency for each type of food. We did not ask for meals timings since the survey was already too long.
  •  
  • Could the content and timing (hours close to bedtime) of TV and games might be more important that the actual time spent?
  • Answer - We asked the tv programs but the data are not yet analyzed. We did not ask the timing since the survey was already too long.
  • A more detailed classification of co-inhabitance, e.g “alone”, “married”, “with children”, “single parent with children”, and “co-inhabiting with non-relative”, would be more informative than civil state, particularly during a mandatory house confinement and social isolation, as that during the lockdown.
  • Answer: These data are included in the present version. The intention was to present them with other housing and geographical circumstances, currently in preparation.

Reviewer 2 Report

Thank you for the opportunity to review this manuscript. The aim of this study is evaluation of sleep and awakening quality during COVID-19 in a large and diversified population in order to identify significant associations and risks in terms of demography, health and health related behaviors, sleep variables, mental health and attitudes. This article is very interesting. Sleep becomes particularly relevant when facing stressing situations like the ones faced during the COVID-19 pandemics. However, some items need to be modified or further clarified.

  • A limitation of the study is of methodological type: including COVID19 positive subjects is a mistake because the possible effects of the virus on sleep are not taken into account as an expression of damage to the CNS.
  • The subgroup of subjects affected by COVID 19 is not discussed in the discussion. Analyze it and discuss it as a subgroup.
  • The conclusions must be expanded and modified in the light of the above
  • It would be usefull to consult the following paper: Partinen M, Bjorvatn B, Holzinger B, Chung F, Penzel T, Espie CA, Morin CM; ICOSS-collaboration group. Sleep and circadian problems during the coronavirus disease 2019 (COVID-19) pandemic: the International COVID-19 Sleep Study (ICOSS). J Sleep Res. 2021 Feb;30(1):e13206. doi: 10.1111/jsr.13206. Epub 2020 Nov 12. PMID: 33179820.

Author Response

Thank you very much for your comments which were effectively applied in a 3rd wave survey currently on line.

The answers to your specific comments are as follows 

  • The subgroup of subjects affected by COVID 19 is not discussed in the discussion. Analyze it and discuss it as a subgroup.
  • Answer- The number of people infected by COVID at this period was very small 122;  30 of which were asymptomatic and 81 had mild symptoms. These data are included in the text. and the chi2 coomparisons with SQ and AQ were included in the results and discussion.
  • The conclusions must be expanded and modified in the light of the above done
  • Answer - It was done

It would be usefull to consult the following paper: Partinen M, Bjorvatn B, Holzinger B, Chung F, Penzel T, Espie CA, Morin CM; ICOSS-collaboration group. Sleep and circadian problems during the coronavirus disease 2019 (COVID-19) pandemic: the International COVID-19 Sleep Study (ICOSS). J Sleep Res. 2021 Feb;30(1):e13206. doi: 10.1111/jsr.13206. Epub 2020 Nov 12. PMID: 33179820.

  • The subgroup of subjects affected by COVID 19 is not discussed in the discussion. Analyze it and discuss it as a subgroup.The number of people affected by COVID at this period was vey small 122, 80 of which were asymptomatic. These data are included in the text.
  • The conclusions must be expanded and modified in the light of the above done

It would be usefull to consult the following paper: Partinen M, Bjorvatn B, Holzinger B, Chung F, Penzel T, Espie CA, Morin CM; ICOSS-collaboration group. Sleep and circadian problems during the coronavirus disease 2019 (COVID-19) pandemic: the International COVID-19 Sleep Study (ICOSS). J Sleep Res. 2021 Feb;30(1):e13206. doi: 10.1111/jsr.13206. Epub 2020 Nov 12. PMID: 33179820.

Answer - The reference was introduced and discussed

Thanks again for the comments

Reviewer 3 Report

Reviewer comments and suggestions

The current article discussed the sleep and awakening quality during COVID19 in a population-based study. The authors revealed these qualities with several parameters and risks based on health-related behaviors, sleep variables, mental health and attitudes. 

The material method chooses the online surveys included 5479 individuals from the general population, sleep disorder patients, COVID involved (MD and Nurses) and COVID affected professionals (Teachers, Psychologists, Dentists). The result section discussed the SQ and AQ were worse in adults, females and high education subjects. Moreover, they also included that feeling worse, having economic problems, depression, anxiety, irritability, and a high Calamity Experience Check List (CECL) during COVID were significantly associated with poor SQ and AQ. 

In conclusion, the authors reported that SQ logistic regression showed gender, morbidities, CECL and awakenings as relevant, whereas for AQ relevant variables further included age and physical activity.

The manuscript needs to be revised based on the below comments as in the current form it cannot be accepted. 

  1. The first three lines of abstract “Big sentence, better to make it short in two small sentences with a clear cut meaning”
  2. Line 44 Age and sex should be mentioned
  3. Line 45-47 The paper needs to well describe method used in the study, was they used an approved online questionnaire?
  4. Line 62-63 The first sentence should be simple, as it shows grammatical mistake
  5. Line 78, You need to start with some paper and then follow your paper
  6. Line 80 characterized their sleep as bad “ what does it mean)
  7. line 83-84 This might suggest that some cultural differences might also affect subjective SQ. (please explain)
  8. Please check line 100 and 110
  9. Line 112-114 reduce into simple and short
  10. Line 156-157 explain it more
  11. Need to explain the term before COVID 19
  12. Table 3 Is these questions were valid previously for SLEEP QUALITY
  13. Line 213 You need to define it
  14. Is there any variable you measure objectively? I mean understand it's an online.
  15. Line 223 need to define (Risk behaviors)
  16. Line 248 bmi is a measurement of obesity, not a total health
  17. The study would be better if the true association was observed in the general population, mix population always provide confusing relationship.
  18. No need to write like this (Line 276-278)
  19. studies need to add more references (line 295) 
  20. Line 300 You have used sleep, disordered patients, as well, how can you define whether they were normal previously and the sleep-disordered patients were due to COVID only
  21. Line 317 Any solid reason for this
  22. In the discussion, the author has to discuss VAS model 
  23. Line 335-336 The sentence need to rewrite
  24. It is better to include table number in the discussion so that reader can follow your paper
  25. Line 359 You can add some references when a new title has been discussed
  26. Please avoid big sentences. (line 373-376)
  27. All references should be changed according to the journal guidelines.

Author Response

Thanks for your review . It was a pleasure to change the manuscript according to your comments.

Below you may find what was done: the answers are enhanced in bold/italic in accordance with your comments. They are often telegraphic for sake of simplicity.

  1. The first three lines of abstract “Big sentence, better to make it short in two small sentences with a clear cut meaning” Done
  2. Line 44 Age and sex should be mentioned Done
  3. Line 45-47 The paper needs to well describe method used in the study, was they used an approved online questionnaire? Done
  4. Line 62-63 The first sentence should be simple, as it shows grammatical mistake Done
  5. Line 78, You need to start with some paper and then follow your paper I do not understand
  6. Line 80 characterized their sleep as bad “ what does it mean) The authors themselves (Trakada et al 2020) classify sleep quality as “good” “average” and “bad”
  7. line 83-84 This might suggest that some cultural differences might also affect subjective SQ. (please explain) done references added
  8. Please check line 100 and 110 done
  9. Line 112-114 reduce into simple and short done
  10. Line 156-157 explain it more done
  11. Need to explain the term before COVID 19 done
  12. Table 3 Is these questions were valid previously for SLEEP QUALITY (the detailed comparison of Sleep Quality prior and during COVID is beyond the aim of this article; it is currently being prepared for another publication)
  13. Line 213 You need to define it It is explained in the methods as a function of percentiles
  14. Is there any variable you measure objectively? I mean understand it's an online. Not possible to measure objectively in on line anonymous surveys
  15. Line 223 need to define (Risk behaviors) done at the introduction
  16. Line 248 bmi is a measurement of obesity, not a total health (the text and the title of the Table have been changed
  17. The study would be better if the true association was observed in the general population, mix population always provide confusing relationship. (this is being presented in another study in which groups are compared; it will be soon sent for publication)
  18. No need to write like this (Line 276-278) (references referring to PSQI in work days omitted)
  19. studies need to add more references (line 295) done
  20. Line 300 You have used sleep, disordered patients, as well, how can you define whether they were normal previously and the sleep-disordered patients were due to COVID only We are not stating that they were “normal” before, we are evaluating the individual differences prior and during COVID19; in such situation it does not matter whether they are patients or not
  21. Line 317 Any solid reason for this In the text the number of persons at the household is in line with this statement
  22. In the discussion, the author has to discuss VAS model it is discussed 
  23. Line 335-336 The sentence need to rewrite done
  24. It is better to include table number in the discussion so that reader can follow your paper Not comfortable doing it
  25. Line 359 You can add some references when a new title has been discussed done
  26. Please avoid big sentences. (line 373-376)done
  27. All references should be changed according to the journal guidelines.done

Thank you again

Round 2

Reviewer 1 Report

I thank the authors for their effort to respond to my previous comments. Most points have been answered, and I understand the particularities of the study mentioned in their responses (e.g. lack of some details due to length of survey).

Nevertheless, a further elaboration on what makes the use of AQ scale in this manuscript a novelty of the study, and objectively better that previous scales used, is still required. The authors claim to already have discussed this comparison (presumably in lines 73-77, since they did not explicitly mention where their response/addition is located in the new version), however their response consists only of two sentences, one for each given reference, explaining what the SSA is and its moderate correlation with objective measurements. Although I do agree with the authors that the AQ scale is indeed important, and their response is on the right direction, they should also point out any methodological differences between their AQ scale and SSA-2 scale. In other words, what makes their AQ scale representative, when the SSA-2 scale was not highly correlated with objective sleep parameters? If the main difference is the implementation of the 1-10 VAS (as mentioned in lines 315-318), compared to the 4 levels of the SSA-2, then please add your response there (after line 320). In general, the response to this question should rather be located in the Discussion section, (~line 321), where the VAS, AQ scale and their importance are discussed, and not in the introduction, where it mostly confuses.

Regarding the CECL citation, I trust the authors that the study explaining/evaluating their calamity scale will be published beforehand, so the reader will be able to understand better the correlations/conclusions drawn in the current manuscript, and how representative these are. 

I thank the authors again for their time to consider my suggestions.

Author Response

Thanks again for the comments. I hope you agree with the changes in this review.

Comments concerning Awakening Quality.

The authors are not comparing standard validated questionnaires for SQ and AQ evaluation, such as the PSQI, Jenkins, SSA etc. with the VAS used. Such comparisons are out of the scope of the present publication. The aim of the present publication was to gather quantitative, reliable and easy to get information in a quite long online survey.

The SSA inclusion in lines 73-77 of the introduction follows the same reasoning used for the PSQI, Jenkins, MOSS, ISS etc. These questionnaires are not discussed only referred.

At the discussion, line 309 the following sentence was included "while AQ evaluation used the SSA questionnaire [8,9]"

At the discussion, line 322 the sentence “is in a sense a novelty of this work” was omitted and references 8 and 9 were included in the subsequent line 323.

Comments concerning Tomé et al in publication

This article was not included in the references. It is mentioned in the text and it is effectively in publication

Reviewer 3 Report

No more comments

Author Response

With our thanks and regards no changes were required